# Manifold Learning and Alignment with Generative Adversarial Networks

## Abstract

We present a generative adversarial network (GAN) that conducts manifold learning and alignment (MLA): A task to learn the multi-manifold structure underlying data and to align those manifolds without any correspondence information. Our main idea is to exploit the powerful abstraction ability of encoder architecture. Specifically, we define multiple generators to model multiple manifolds, but in a particular way that their inverse maps can be commonly represented by a single smooth encoder. Then, the abstraction ability of the encoder enforces semantic similarities between the generators and gives a plausibly aligned embedding in the latent space. In experiments with MNIST, 3D-Chair, and UT-Zap50k datasets, we demonstrate the superiority of our model in learning the manifolds by FID scores and in aligning the manifolds by disentanglement scores. Furthermore, by virtue of the abstractive modeling, we show that our model can generate data from an untrained manifold, which is unique to our model.

## 1 Introduction

Real-world datasets generally have multi-manifold structure. Smoothly varying features such as the size or the pose of an animal introduce smooth manifold structure, while the existence of discrete features such as species makes the structure not a single manifold but multiple disconnected manifolds (e.g., one manifold per species) (Khayatkhoei et al., 2018). Importantly, these manifolds are correlated to each other in their structure since the transformation rules according to the size (stretching the foreground patch) or the pose (rotating the patches of body parts) are similar even for different species. Hence, our desire is to build a model that learns not only the multi-manifold structure but also the correlations between the manifolds.

As for the learning of manifold structure, generative adversarial networks (GANs) (Goodfellow et al., 2014) have been known for their remarkable performance. In particular, recently proposed GAN models have successfully demonstrated multi-manifold learning by using multiple generator networks (Khayatkhoei et al., 2018; Ghosh et al., 2017; Hoang et al., 2018) or giving a mixture density on the latent space (Xiao et al., 2018; Gurumurthy et al., 2017). However, none of these models have taken the correlations between the manifolds into account.

On the other hand, a field of research named *manifold alignment* (Ham et al., 2003) has its primary concern in learning the correlations between the manifolds. In particular, unsupervised manifold alignment (Cui et al., 2014; Wang & Mahadevan, 2009) aims to learn plausible correspondences between the data points of different manifolds, without any ground-truth correspondence information is given. However, most existing methods focus only on finding the correspondences, not on learning the structure of the manifolds itself.

In this work, we propose a model that performs both of the tasks, which we call as *manifold learning and aligning GAN* (MLA-GAN). MLA-GAN is similar to the GANs with multiple generators (Khayatkhoei et al., 2018; Ghosh et al., 2017; Hoang et al., 2018), but the generators are restricted such that their inverse maps can be represented by a single deep encoder. Due to the abstracting property of deep encoders, this restriction guides the inverse maps to learn semantically similar mappings, and the model achieves a semantically plausible manifold alignment. The main contributions of this work can be summarized as follows.

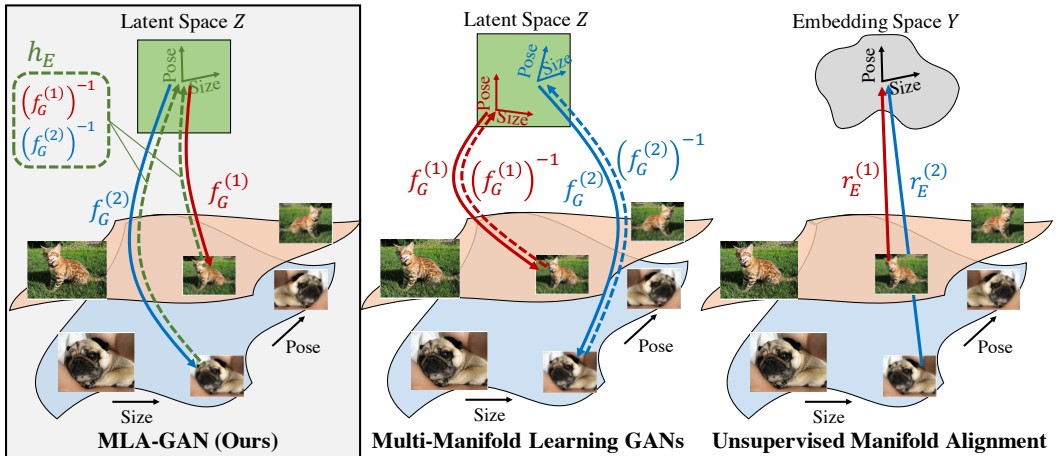

Figure 1: The proposing MLA-GAN model is illustrated along with multi-manifold learning GANs (abbr. mmGANs) and unsupervised manifold alignment methods (abbr. UMA). In **MLA-GAN**, the latent features of cats and dogs manifolds are aligned. This is achieved by restricting the inverses of the generative maps $(f^{(1)})^{-1}$, $(f^{(2)})^{-1}$ to be represented by a common encoder $h_E$ (see Eq. 1). In **mmGANs**, the latent features are not aligned as $f^{(1)}$ and $f^{(2)}$ do not share any structures. In **UMA**, the embedding coordinates are trained to be aligned, but as $r_E^{(1)}$, $r_E^{(2)}$ are linear maps, they lose information about the structure of the manifolds.

- **The MLA problem:** Up to our knowledge, MLA-GAN is the first model that performs both multi-manifold learning and alignment with real-world datasets. It shows state-of-the-art FID scores in data generation and plausible alignments with disentangled features.

- **Generalization to an untrained manifold:** Due to the abstracting nature of an encoder, the smooth features can be inferred even for data lying on an untrained manifold. Exploiting this, one can generate other data points lying on this manifold.

- **Easily applicable**: Due to the new regularizer we propose, the inverse maps used for restricting the generators do not need to be actually computed. This makes MLA-GAN easily applicable to any existing GAN architectures.

## 2 MANIFOLD LEARNING AND ALIGNMENT WITH GAN

### 2.1 BACKGROUNDS

**Multi-Manifold Learning GANs**   Conventional GANs (Goodfellow et al., 2014) model data by transforming a latent-space distribution $p(z)$ with a generating map $f_G : Z \to X$. If $f_G$ is smooth and injective[1], the image of $f_G$ is a smooth manifold $M \subset X$, thus the data are modeled as they are lying on this manifold. To deal with multiple manifolds, recent extensions (Khayatkhoei et al., 2018; Ghosh et al., 2017; Hoang et al., 2018) employ multiple generating maps $\{f_G^{(i)}\}_{i=1}^A$ where each $f_G^{(i)}$ accounts for the $i$-th manifold $M^{(i)}$.

**Unsupervised Manifold Alignment**   Most existing methods formulate the manifold alignment problem as finding embeddings of manifolds into a common low-dimensional space $Y$. This set of embeddings $\{r_E^{(i)} : M^{(i)} \to Y\}_i$ essentially defines an alignment of manifolds; data points that are mapped to the same point in $Y$ correspond to each other. To achieve a plausible alignment, the embeddings are trained to minimize several loss terms guiding geometrical similarity (see Sec. 3).

---

[1]Although these conditions are not guaranteed for general neural networks, they are approximately met or observed in practice (Shao et al., 2017). Additionally, the support of $p(z)$ needs to be a completely separable Hausdorff space, but this condition is met for the usual choices of $p(z)$ (e.g., unimodal Gaussian).

## 2.2 MAIN IDEA

To model multiple manifolds, we use multiple generating maps $\{f_G^{(i)} : Z \to M^{(i)}\}_i$ as with the aforementioned GAN models. Interestingly, the inverses[2] of these maps $\{(f_G^{(i)})^{-1} : M^{(i)} \to Z\}_i$ can be regarded as a set of embeddings (where $Y$ is replaced by $Z$), which means that *the manifold alignment is automatically defined as we define the manifolds.* However, this alignment is not at all semantically plausible unless properly regulated. Here, we regulate it by restricting the generating maps such that their inverses to be commonly represented by a single encoder $h_E : X \to Z$. Formally, we restrict the generating maps as follows:

$$\{f_G^{(i)}\}_i : \text{A set of functions } (Z \to M^{(i)} \subset X) \ s.t.$$
$$\exists h_E \in \mathcal{H} \text{ satisfying } (f_G^{(i)})^{-1}(x) = h_E(x) \quad \forall i, \forall x \in M^{(i)}. \tag{1}$$

Here, $\mathcal{H}$ is a certain set of functions (to be clearer soon), and $h_E$ being an element of $\mathcal{H}$ is an important constraint. Without this constraint, one can always find $h_E$ satisfying the above condition, for any sets of $\{f_G^{(i)}\}_i$, because $h_E$ is merely a union of functions $\{(f_G^{(i)})^{-1}\}_i$ extending the domain from $\bigcup_i M^{(i)} \subset X$ to $X$. However, if $\mathcal{H}$ is, for example, a set of smooth functions and $h_E \in \mathcal{H}$, then $\{(f_G^{(i)})^{-1}\}_i$ will be restricted to be similar to each other and the alignment will become more plausible. In our model, $\mathcal{H}$ is defined as a *set of functions represented by deep encoder networks.* We explain the intuition behind this design in what follows.

**Intuition** Consider a dataset containing images of cats and dogs. It is not difficult to imagine a deep encoder network that takes an image from this dataset and outputs the size and pose regardless of the species. At least, we could have trained this encoder easily if the size and pose labels were given. Also, even if we train the encoder using the cat data only, we still expect it works fairly well for the dogs due to the powerful abstraction and generalization abilities of deep encoders. We want to exploit these abilities of deep encoders for aligning the manifolds. If we let $\mathcal{H}$ be a set of deep encoders, its element $h_E$, once well-trained, will abstract the species away and outputs the size and pose. This gives a nice manifold alignment, where a cat and a dog images with the same size and pose correspond to each other.

**Remaining Question** Is $\mathcal{H}$ being a set of deep encoders enough to obtain the abstraction and generalization? An answer to this question is related to how deep networks generalize so well despite the large number of parameters they have. Several works conjecture that it is due to the SGD (Zhang et al., 2016; Brutzkus et al., 2018), but this is not a fully developed theory yet. At least, we can assume that deep encoders learn simpler, generalization-friendly maps among others, based on various cases (e.g., VGG-Net (Simonyan & Zisserman, 2014), Fast R-CNN (Girshick, 2015)).

## 2.3 MLA-GAN

What remains is to implement the generating maps with deep neural networks. The main challenge is to make the deep networks comply with the constraint given in Eq. 1. In this section, we explain how this challenge can be achieved in the proposing MLA-GAN model in detail. We first focus on a single linear layer and prove the following (detailed in Proposition 1):

> If linear generating maps are defined with the *same weight* and the *same tangential components of bias,* their inverses can always be represented by a single linear encoder. Thus, the maps comply with Eq. 1.

Then, to realize the above, we introduce a new form of regularizer. After, we move on to the entire network and derive MLA-GAN by applying the same principles as the linear layer.

### 2.3.1 SINGLE LINEAR LAYER

Consider a set of linear generating maps $\{f_G^{(i)} : Z \to M^{(i)} \subset X\}_i$; each of the maps is defined as

$$f_G^{(i)}(z) := U^{(i)} z + a^{(i)},$$

---

[2]With the same assumption as Footnote 1.

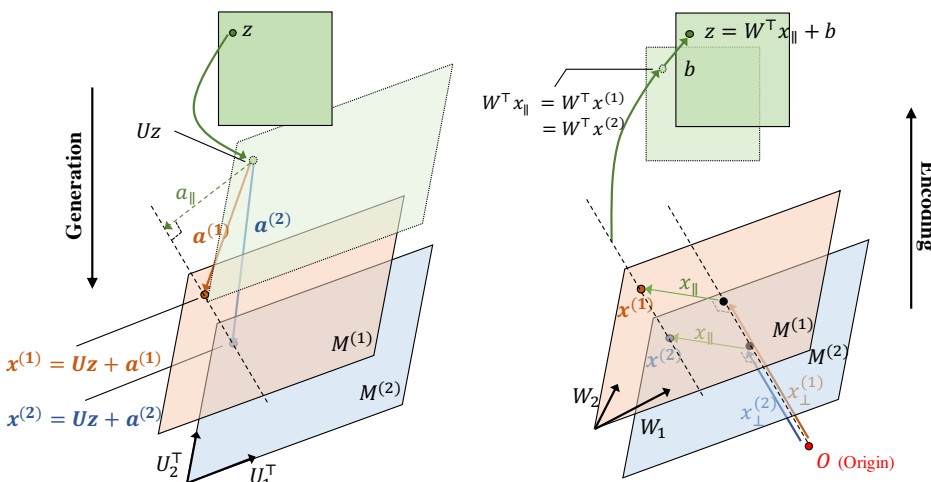

Figure 2: The generators and the encoder of a linear layer are illustrated. **Generation:** A latent point $z$ is transformed by the same weight $U$, then different biases $a^{(1)}$ and $a^{(2)}$ are added to generate $x^{(1)}$ and $x^{(2)}$ for different manifolds. The biases are regularized to have *the same tangential component* (to the column space of $W$ or $U^\top$) $a_\parallel$ to comply with Eq. 1. **Encoding:** The points $x^{(1)}$ and $x^{(2)}$ lying on different manifolds are mapped to the same $z$ if they have the same tangential components $x_\parallel$. Note this becomes false if the biases are not regularized.

where $U^{(i)} \in \mathbb{R}^{d_X \times d_Z}$ ($d_X > d_Z$) and $a^{(i)} \in \mathbb{R}^{d_X}$ are the weight and bias parameters respectively. We assume $U^{(i)}$ is a full-rank matrix (rank-$d_Z$) such that $f_G^{(i)}$ is injective (see Footnote 1). Then, the inverse $(f_G^{(i)})^{-1} : M^{(i)} \to Z$ can be derived as

$$(f_G^{(i)})^{-1} = (U^{(i)})^+ (x^{(i)} - a^{(i)}) \tag{2}$$

where $(U^{(i)})^+ := \left((U^{(i)})^\top U^{(i)}\right)^{-1} (U^{(i)})^\top$ denotes the pseudo-inverse of $U^{(i)}$. As noted in Eq. 1, our desire is to restrict the inverse maps $\{(f_G^{(i)})^{-1}\}_i$ such that they can be represented by a single encoder $h_E : X \to Z$. One simple way to achieve this is to use the following proposition.

**Proposition 1.** *If the linear generating maps $\{f_G^{(i)}(z)\}_i$ are restricted to have the same weight $U$ and to have the same tangential components of bias $a_\parallel$ (i.e., $a^{(i)} = a_\parallel + a_\perp^{(i)}$), then their inverses $\{(f_G^{(i)})^{-1}\}_i$ can be represented by a single linear encoder $h_E(x) := W^\top x + b$ where $W = U(U^\top U)^{-1}$ and $b = -W^\top a_\parallel$.*

*Proof.* Note the bias $a^{(i)}$ can be decomposed into the tangential component $a_\parallel^{(i)}$ and the normal component $a_\perp^{(i)}$ to the column space of $U^\top$. Substituting the condition $a^{(i)} = a_\parallel + a_\perp^{(i)}$ and the same weight $U$, Eq. 2 is written as

$$\begin{aligned}(f_G^{(i)})^{-1} &= U^+ x - U^+ (a_\parallel + a_\perp^{(i)}) \\ &= U^+ x - U^+ a_\parallel \\ &= W^\top x + b. \qquad \square\end{aligned}$$

**Bias Regularizer** When implementing Proposition 1, making $\{U^{(i)}\}_i$ the same is as trivial as setting the same weight $U$ for all $f_G^{(i)}$, but making $\{a_\parallel^{(i)}\}_i$ the same is not trivial since the tangential direction keeps changing while training. One solution would be to use a regularizer minimizing the sum of the variance: $\text{trace}(\text{cov}(a_\parallel^{(i)}))$. However, computing this term is intractable due to the inversion $(U^\top U)^{-1}$ inside of $a_\parallel^{(i)} = U(U^\top U)^{-1} U^\top a^{(i)}$.

**Theorem 1.** *The following inequality holds:*

$$trace\left(cov(U^\top a^{(i)})\right) \geq \frac{1}{d_z} H(\{\lambda_k\}_{k=1}^{d_z}) trace\left(cov(a_\parallel^{(i)})\right)$$

*where $\{\lambda_k\}_{k=1}^{d_z}$ denotes the eigenvalues of $U^\top U$ and $H(\cdot)$ denotes harmonic mean.*

*Proof.* See Appendix B. □

As the harmonic mean in Theorem 1 is constant from the perspective of $a_\parallel^{(i)}$, we can minimize the original term by minimizing the upper bound instead. With an additional log function to match the scale due to the dimensionality, we propose the upper bound as a regularizer to make $a_\parallel^{(i)}$ the same:

$$\textbf{Regularizer:}\quad R_{bias} = \log\left(\text{trace}\left(\text{cov}(U^\top a^{(i)})\right)\right). \tag{3}$$

### 2.3.2 ENTIRE NETWORK

A single linear layer is, of course, insufficient as linear generators can only model the linear manifolds, and a linear encoder can align the manifolds only linearly by removing the normal components. So, we implement the generating maps with deep networks to model complex nonlinear manifolds. As usual, the deep networks are constructed by interweaving linear layers and nonlinear activations, and optionally batch-normalization layers. But, each linear layer here implements the conditions of Proposition 1. That means, for the $l$-th linear layer of each deep generating map $f_G^{(i)}$, their weights are set as the same, denoted as $U_l$, and their biases $\{a_l^{(i)}\}_i$ are regularized to have the same tangential components using Eq. 3.

Note that the above set of deep generating maps comply with Eq. 1. As all the linear layers, nonlinear activations (we use LeakyReLU; see Appendix C), and batch-norm layers are injective, the deep generating maps have inverses. These deep inverse maps can be represented by a single deep encoder, as the inverses of all the component-linear maps can be represented by single encoders.

**Training**   As above, let us denote the weight of the $l$-th linear layer as $U_l$ and the biases as $\{a_l^{(i)}\}_{i=1}^A$. Then, we can express the data generating distribution as an ancestral sampling:

$$x \sim f_G^{(i)}\left(z; \{U_l, a_l^{(i)}\}_{l=1}^L\right) \quad \text{where } z \sim p(z),\ i \sim \pi_i.$$

Here, $\pi_i$ stands for the probability of selecting the $i$-th bias. This probability could be also learned using the method proposed in Khayatkhoei et al. (2018), but it is beyond our scope and we fix it as $1/A$. Now, denoting the real data distribution as $p_R$ and the fake distribution that the above sampling presents as $p_G$, we define our GAN losses as:

$$\textbf{MLA-GAN Loss:}\quad \begin{cases} \mathcal{L}_G = -E_{x \sim p_G}[D(x)] + \lambda \sum_{l=1}^L \log\left(\text{trace}\left(\text{cov}(U_l^\top a_l^{(i)})\right)\right) \\ \mathcal{L}_D = E_{x \sim p_G}[D(x)] - E_{x \sim p_R}[D(x)] \end{cases}$$

where $\mathcal{L}_G$ and $\mathcal{L}_D$ are the generator and the discriminator losses respectively and $\lambda$ is a regularization weight. We use Wasserstein GAN (WGAN) Arjovsky et al. (2017) so the discriminator $D(x)$ is restricted to a $k$-Lipschitz function.

**Encoding**   Due to the bias regularizer, we do not need to concretize the encoder $h_E$ by inverting $f_G^{(i)}$ during training. But, when encoding is needed, we can obtain the latent codes and biases by minimizing the distance along with the bias regularizer as follows (as suggested in Ma et al. (2018)).

$$z, \{a_l\}_{l=1}^L = \arg\min_{\tilde{z}, \{\tilde{a}_l\}_{l=1}^L} \left\|x - f_G\left(\tilde{z}; \{\tilde{a}_l\}_{l=1}^L\right)\right\|^2 + \mu \sum_{l=1}^L \log\left(\left\|U_l^\top a_l - U_l^\top \bar{a}_{l,\parallel}\right\|^2\right), \tag{4}$$

Importantly, one can encode an image that is considerably different from the trained images using this method. By virtue of the abstraction ability of the encoder, the encoded latent code is actually semantically plausible. Moreover, as the encoded biases fully describe the generating map, one can virtually generate any data lying on the same manifold. This is unique to our model and to be further discussed in Sec. 4.3.

Table 1: FID (smaller is better) and Disentanglement (larger is better) scores are shown. We compare WGAN (Arjovsky et al., 2017), DMWGAN (Khayatkhoei et al., 2018), $\beta$-VAE (Higgins et al., 2016), InfoGAN (Chen et al., 2016) with our model. The mean and std. values are computed from 10 (MNIST) and 5 (3D-Chair) replicated experiments.

| | | WGAN | DMWGAN | $\beta$-VAE | InfoGAN | MLA-GAN (Ours) | MLA-GAN, $\lambda = 0$ |
|---|---|---|---|---|---|---|---|
| FID | MNIST | $10.13 \pm 3.16$ | $\mathbf{5.41 \pm 0.34}$ | $58.43 \pm 0.23$ | $12.17 \pm 1.30$ | $\mathbf{5.69 \pm 0.89}$ | $15.74 \pm 10.00$ |
| | 3D-Chair | $\mathbf{125.32 \pm 1.16}$ | $184.5 \pm 31.5$ | $217.12 \pm 0.55$ | $187.94 \pm 9.51$ | $\mathbf{125.27 \pm 4.34}$ | $128.44 \pm 7.06$ |
| Disent. | MNIST (slant) | $1.62 \pm 0.41$ | $1.08 \pm 0.04$ | $\mathbf{5.04 \pm 1.19}$ | $1.24 \pm 0.15$ | $2.15 \pm 0.17$ | $1.76 \pm 0.35$ |
| | MNIST (width) | $1.68 \pm 0.49$ | $1.11 \pm 0.06$ | $\mathbf{5.63 \pm 0.75}$ | $1.18 \pm 0.11$ | $2.93 \pm 0.60$ | $2.75 \pm 0.67$ |
| | 3D-Chair (height) | $2.14 \pm 0.20$ | $1.14 \pm 0.05$ | $\mathbf{8.10 \pm 0.20}$ | $1.41 \pm 0.34$ | $3.27 \pm 1.73$ | $2.76 \pm 0.31$ |
| | 3D-Chair (bright.) | $3.53 \pm 0.80$ | $1.20 \pm 0.14$ | $3.96 \pm 0.20$ | $3.02 \pm 0.88$ | $\mathbf{4.45 \pm 0.66}$ | $4.24 \pm 0.54$ |

## 3 RELATED WORK

**Multi-Manifold Learning GANs**   There exist roughly three different methods to handle multiple manifolds with GANs (see also Appendix A). Using multiple generators (DMWGAN (Khayatkhoei et al., 2018); Ghosh et al. (2017); Hoang et al. (2018)) is the simplest one, but it cannot align the manifolds as the generators are independent to each other without any regulations. Giving a mixture density for the latent space prior (Xiao et al., 2018; Gurumurthy et al., 2017) is another method; although this method uses a single generating map, it hardly aligns the manifolds since the manifolds are modeled by different regions in the latent space where the mappings are effectively different from each other. The last method (InfoGAN (Chen et al., 2016)) introduces discrete latent variables and concatenate them with the original continuous latent variables. Here, the discrete information is slowly blended in, layer by layer, and modulates the generating map. As the overall structure is shared among the manifolds to some degree, this method does align the manifolds; but as it is not explicitly guided, the performance is not as good as MLA-GAN (Table 1).

**Unsupervised Manifold Alignment**   The existing methods (Cui et al., 2014; Wang & Mahadevan, 2009) find a correspondence between the points of different manifolds by learning embeddings, with consideration for geometrical similarity. As the geometry is captured by adjacency matrices, one shortcoming is that the alignment can be done only in a point-to-point manner. Moreover, as the embeddings are modeled as linear functions, it is inevitable to lose information about the manifolds.

**Deep generative models involving an encoder**   Utilization of an encoder is a notable feature of MLA-GAN, but there are several other deep generative models using encoders, such as adversarial auto-encoders (Makhzani et al., 2015), variational auto-encoders (Kingma & Welling, 2013; Higgins et al., 2016) and BiGAN (Donahue et al., 2016). However, there is a large difference: The encoders of these models capture all the necessary information in order to reconstruct the data, while the encoder of MLA-GAN removes the discrete information in the latent space to abstract the manifolds.

## 4 EXPERIMENTS

**Datasets**   We experiment on *MNIST* (Lecun et al., 1998), *3D-Chair* (Aubry et al., 2014) and *UT-Zap50k* (Yu & Grauman, 2014) image datasets. 3D-Chair contains 1393 distinct chairs rendered from 62 different viewing angles (total 86,366 images); in experiments, only front-looking 44,576 images are used and rescaled to 64x64 grayscale images. UT-Zap50k contains images of 4 different types of shoes (total 50,025 images); the images are rescaled to 32x32.

**Model Architecture**   We use DCGAN (Radford et al., 2015)-like model architectures for all the datasets (see Appendix C for the complete information). For each of the linear layers in the generator, the number of biases $A$ is set as 10 (MNIST), 20 (3D-Chair), and 4 (UT-Zap50k). Although our multi-biased linear layer can be applied to both fully-connected and transposed-convolution layers, we apply it only to the former. This is sufficient for our purpose since discrete features rarely exist for such small-sized kernel patches. In the discriminator, we use spectral normalization (Miyato et al., 2018) to achieve the $k$-Lipschitz condition for WGAN. For training and encoding, Adam (Kingma & Ba, 2014) is used with the defaults except for the learning rate, 0.0002.

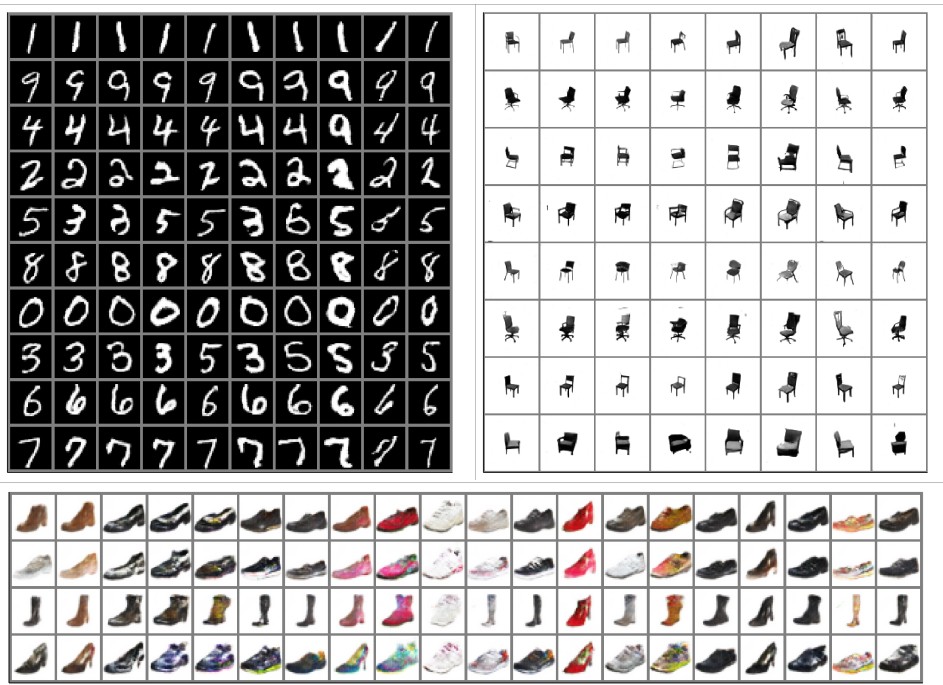

Figure 3: Images generated from the trained MLA-GANs. Each $i$-th row presents samples from the $i$-th manifold (only 8 out of 20 are shown for 3D-Chair) and each column presents the samples generated from the same latent code $z$, which is randomly sampled from $p(z)$. See also Appx. J.

### 4.1 MULTI-MANIFOLD LEARNING

MLA-GAN shows great performance in multi-manifold learning and alignment, both qualitatively and quantitatively. Looking at Fig. 3 row-wise, we can see that MLA-GAN learns distinct manifolds well, where each manifold accounts for a different discrete class: e.g., different digits; rolling vs. armchairs; boots vs. flat shoes. Column-wise, we can see that the smooth features are well aligned among the manifolds: e.g., stroke weight, the slant of numbers; viewing angle of chairs; colors of shoes. Note that these alignments are qualitatively much better than that of the existing alignment methods given in Fig. D.2. For quantitative evaluation, we compute FID scores (Heusel et al., 2017), which is widely used to measure the diversity and quality of the generated image samples, reflecting the manifold learning performance overall. Table 1 shows that our model gives better or comparable scores than others.

### 4.2 MANIFOLD ALIGNMENT AND DISENTANGLED FEATURES

To further investigate the manifold alignment performance, we examine how much the learned latent features are disentangled. We take a few images from the dataset and manually change one of the smooth features that corresponds to a known transformation (e.g., sheer transform). Then, we encode these images to the latent codes using our model, analyze the principal direction of the change, and compute the linearity of the change as the disentanglement score (see Appendix E). Fig. 4 shows that the learned smooth features are well disentangled along the principal changing directions. Table 1 shows that our model gets better scores than other models, but sometimes not as good as $\beta$-VAE. However, it should be noted that the FID scores of $\beta$-VAE is significantly worse.

### 4.3 STYLE TRANSFER

We demonstrate a style-transfer between images using our model by matching the latent features (Fig. 5). Specifically, we encode the source and target images, then use the inferred biases of the source and the inferred latent codes of the target to generate the style-transferred image.

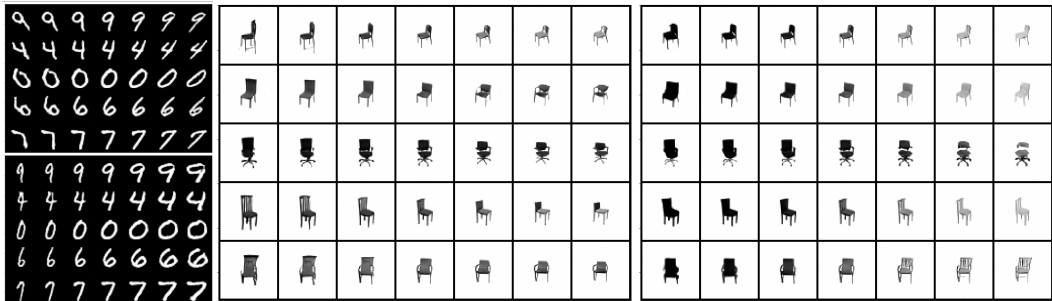

Figure 4: Disentangled features. Images are arranged the same as Fig. 3, except the columns show linear changes in the latent space along the first eigenvector of the disentanglement score analysis (see Sec. 4.2). Slant, width (MNIST), height and brightness (3D-Chair) components are shown.

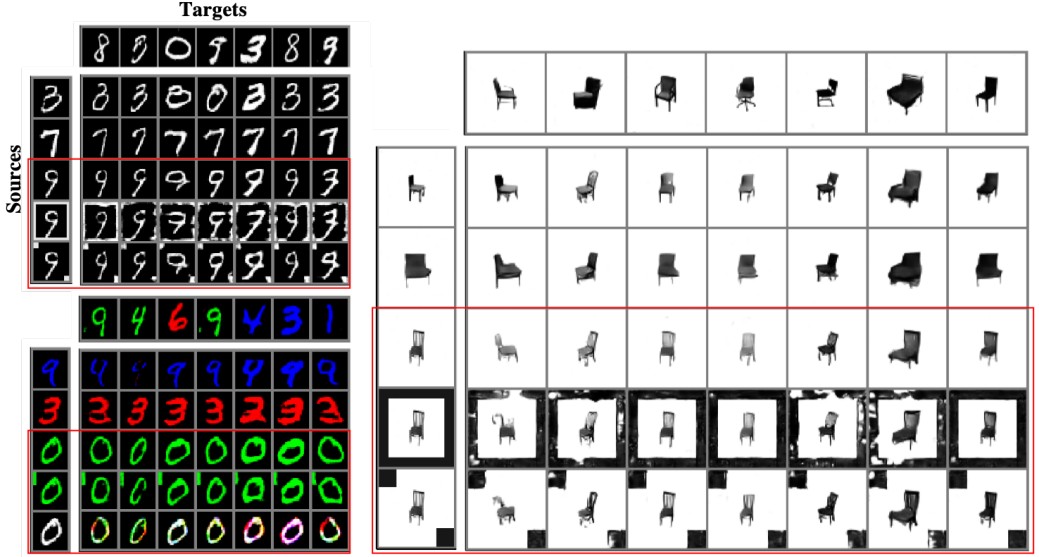

Figure 5: Style transfer results[3]. In each source image set, the fourth and the fifth images are derived from the third image by adding frames, rectangles or changing colors. The derived source images are clearly off the trained distribution, but their style-transfer results are good and consistent with the original style-transfer results (highlighted in red boxes).

Interestingly, MLA-GAN is even able to style-transfer the images that are off the trained distribution (see the red boxes). This is an exclusive property of MLA-GAN due to the encoder. If the encoder generalizes well (which apparently does), it can recognize the smooth features of an image even if the image has noises such as added frames or color modification. Once the smooth features are recognized, the model knows how to transform this image smoothly, similar to the original data, but on this new manifold involving noises (as described in Eq. 4 and below).

## 5 CONCLUSION

We proposed MLA-GAN that performs multi-manifold learning and alignment. To our knowledge, MLA-GAN is the first model that performs both of the tasks for real-world datasets, and it mostly showed superior performance. Unique to MLA-GAN is that it utilizes the abstraction ability of an encoder, from which it showed its potential to infer the untrained manifolds in the style-transfer experiment. If it is trained with larger datasets such as ImageNet in the future, we expect MLA-GAN would become a more versatile tool that can infer the manifold structure of general images.

---

[3]Here, we also train on RGB-MNIST dataset: a simple extension of MNIST by coloring with (R, G, or B).

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

# Appendix

## A    COMPARISON OF VARIOUS GAN MODELS

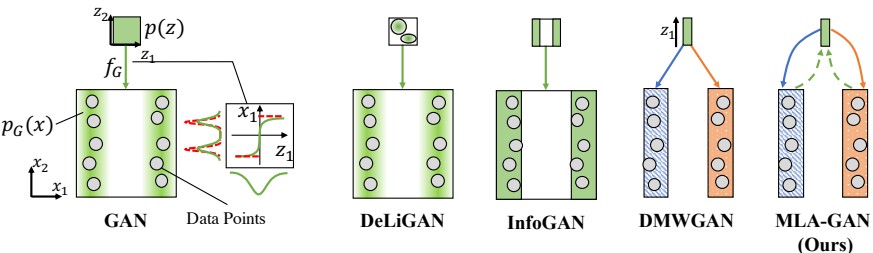

Figure A.1: Various GAN models trained on 2-manifold data are shown schematically. **Conventional GAN** (Goodfellow et al., 2014) learns a highly nonlinear mapping to compensate for the disconnected region with small density. But it is still not perfect compared to the true disconnection (*red-dashed*). **DeLiGAN** (Gurumurthy et al., 2017) and **InfoGAN** (Chen et al., 2016) use a non-connected topology in the latent space. **DMWGAN** (Khayatkhoei et al., 2018) uses distinct generating maps for each manifold. **MLA-GAN (Ours)** is similar but regulated that the inverses of the generating maps can be represented by a common single encoder.

## B    THE PROOF OF THEOREM 1

**Theorem 1.** *The following inequality holds*

$$trace\left(cov(U^\top a^{(i)})\right) \geq \frac{1}{d_z} H(\{\lambda_k\}_{k=1}^{d_z}) trace\left(cov(a_\parallel^{(i)})\right)$$

*where $\{\lambda_k\}_{k=1}^{d_z}$ are the eigenvalues of $U^\top U$ and $H(\cdot)$ denotes a harmonic mean.*

*Proof.* Note that

$$
\begin{aligned}
\text{trace}\left(\text{cov}(a_\parallel^{(i)})\right) &= \text{trace}\left(\text{cov}(U(U^\top U)^{-1} U^\top a^{(i)})\right) \\
&= \text{trace}\left(\frac{1}{A-1}\sum_{i=1}^{A} U(U^\top U)^{-1}U^\top(a^{(i)}-\bar{a})(a^{(i)}-\bar{a})^\top U(U^\top U)^{-1}U^\top\right) \\
&= \text{trace}\left(\frac{1}{A-1}\sum_{i=1}^{A} U^\top(a^{(i)}-\bar{a})(a^{(i)}-\bar{a})^\top U(U^\top U)^{-1}\right) \\
&= \text{trace}\left(\text{cov}(U^\top a^{(i)})(U^\top U)^{-1}\right) \\
&\leq \text{trace}\left(\text{cov}(U^\top a^{(i)})\right)\text{trace}\left((U^\top U)^{-1}\right),
\end{aligned}
$$

where the second and the fourth lines use the definition of the covariance, third line is obtained from the cyclic property of trace and the last line is obtained from the Cauchy-Schwarz inequality of the positive semi-definite matrices. Thus,

$$
\begin{aligned}
\text{trace}\left(\text{cov}(U^\top a^{(i)})\right) &\geq \text{trace}\left(\text{cov}(a_\parallel^{(i)})\right)/\text{trace}\left((U^\top U)^{-1}\right) \\
&= \frac{1}{D} H(\{\lambda_d\}_{d=1}^{D})\text{trace}\left(\text{Var}(a_\parallel^{(i)})\right)
\end{aligned}
$$

where $\{\lambda_d\}_{d=1}^{D}$ are the eigenvalues of $U^\top U$ and $H(\cdot)$ denotes the harmonic mean. $\qquad\square$

## C  MODEL ARCHITECTURE AND EXPERIMENTING ENVIRONMENTS

We used machines with one NVIDIA Titan Xp for the training and the inference of all the models.

### C.1  MNIST

We use $A = 10$ distinct decoding biases in the model. In the training, we set the regularization weight $\lambda = 0.05$ and use the Adam optimizer with learning rate 0.0002. In the encoding, we use the Adam optimizer with learning rate 0.1, and the set the regularization weight $\mu = 0.1$.

Table C.1: MLA-GAN architecture used for MNIST dataset

| Generator | Discriminator |
|---|---|
| Input(8) | Input(1,28,28) |
| Full(1024), BN, LReLU(0.2) | Conv(c=64, k=4, s=2, p=1), BN, LReLU(0.2) |
| Full(6272), BN, LReLU(0.2) | Conv(c=128, k=4, s=2, p=1), BN, LReLU(0.2) |
| ReshapeTo(128,7,7) | ReshapeTo(6272) |
| ConvTrs(c=64, k=4, s=2, p=1), BN, LReLU(0.2) | Full(1024), BN, LReLU(0.2) |
| ConvTrs(c=32, k=4, s=2, p=1), BN, LReLU(0.2) | Full(1) |
| ConvTrs(c=1, k=3, s=1, p=1), Tanh | |

#### C.1.1  NOTES ON THE OTHER COMPARED MODELS

Overall, we match the architecture of other models with our model for fair comparison. Some differences to note are:

- **DMWGAN**: We used 10 generators. Each generator has the same architecture as ours except the number of features or the channels are divided by 4, to match the number of trainable parameters. Note that 4 is the suggested number from the original paper.

- $\beta$-**VAE**: We used Bernoulli likelihood.

- **InfoGAN**: Latent dimensions consist of 1 discrete variable (10 categories), 2 continuous variables and 8 noise variables.

### C.2  3D-CHAIR

We use $A = 20$ distinct decoding biases in the model. In the training, we set the regularization weight $\lambda = 0.05$ and use the Adam optimizer with learning rate 0.0002. In the encoding, we use the Adam optimizer with learning rate 0.1, and the set the regularization weight $\mu = 0.1$.

Table C.2: MLA-GAN architecture used for 3D-Chair dataset.

| Generator | Discriminator |
|---|---|
| Input(10) | Input(1,64,64) |
| Full(256), BN, LReLU(0.2) | Conv(c=64, k=4, s=2, p=1), BN, LReLU(0.2) |
| Full(8192), BN, LReLU(0.2) | Conv(c=128, k=4, s=2, p=1), BN, LReLU(0.2) |
| ReshapeTo(128,8,8) | Conv(c=128, k=4, s=2, p=1), BN, LReLU(0.2) |
| ConvTrs(c=64, k=4, s=2, p=1), BN, LReLU(0.2) | ReshapeTo(8192) |
| ConvTrs(c=32, k=4, s=2, p=1), BN, LReLU(0.2) | Full(1024), BN, LReLU(0.2) |
| ConvTrs(c=16, k=4, s=2, p=1), BN, LReLU(0.2) | Full(1) |
| ConvTrs(c=1, k=3, s=1, p=1), Tanh | |

### C.2.1 NOTES ON THE OTHER COMPARED MODELS

- **DMWGAN**: We used 20 generators. Each generator has the same architecture as ours except the BatchNorms are removed and the number of features or the channels are divided by 4, to match the number of trainable parameters. Note that 4 is the suggested number from the original paper. Note that this was the best setting among what we have tried (division number 2; ones with the BatchNorms).

- $\beta$-**VAE**: We used Bernoulli likelihood.

- **InfoGAN**: Latent dimensions consist of 3 discrete variables (20 categories), 1 continuous variable and 10 noise variables.

### C.3 UT-ZAP50K

We use $A = 4$ distinct decoding biases in the model. For the regularization weight in the training, we start with $\lambda = 5e - 6$ then raise to $\lambda = 5e - 4$ after 300 epochs.

Table C.3: MLA-GAN architecture used for UT-Zap50k dataset.

| Generator | Discriminator |
|---|---|
| Input(8) | Input(3,32,32) |
| Full(512), BN, LReLU(0.2) | Conv(c=128, k=4, s=2, p=1), BN, LReLU(0.2) |
| Full(1024), BN, LReLU(0.2) | Conv(c=256, k=4, s=2, p=1), BN, LReLU(0.2) |
| Full(8192), BN, LReLU(0.2) | Conv(c=512, k=4, s=2, p=1), BN, LReLU(0.2) |
| ReshapeTo(512,4,4) | ReshapeTo(8192) |
| ConvTrs(c=256, k=4, s=2, p=1), BN, LReLU(0.2) | Full(1024), BN, LReLU(0.2) |
| ConvTrs(c=128, k=4, s=2, p=1), BN, LReLU(0.2) | Full(512), BN, LReLU(0.2) |
| ConvTrs(c=64, k=4, s=2, p=1), BN, LReLU(0.2) | Full(1) |
| ConvTrs(c=3, k=3, s=1, p=1), Tanh | |

## D MANIFOLD ALIGNMENT RESULTS OF THE EXISTING METHODS

We show the manifold alignment results of Wang & Mahadevan (2009) for MNIST dataset.

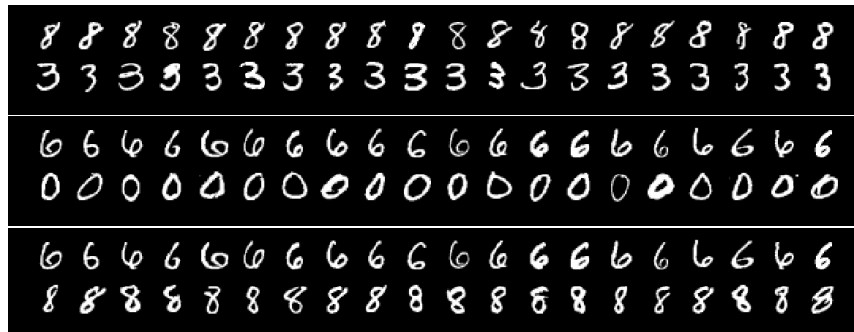

Figure D.2: The manifold alignment results of Wang & Mahadevan (2009) for MNIST dataset.

# E    DISENTANGLEMENT SCORE

To compute the disentanglement score, we first take 500 images from the dataset and manually change one of the smooth features that corresponds to a known transformation. For example, we change the slant of the MNIST digits by taking a sheer transform. With 11 different degrees of the transformation, we obtain 5500 transformed images in total. We encode these images to obtain the corresponding latent codes and subtract the mean for each group of the images (originates from the same image) to align all the latent codes. Then, we conduct Principal Component Analysis (PCA) to obtain the principal direction and the spectrum of variations of the latent codes. If the latent features are well disentangled, the dimensionality of the variation should be close to one. To quantify how much it is close to one, we compute the ratio of the first eigenvalue to the second eigenvalue of the PCA covariance, and set it as the disentanglement score.

# F    EFFECT OF THE NUMBER OF GENERATORS, $A$

To investigate the effect of the number of generators (or biases), $A$, we train our model on MNIST with different $A$ values then compute the FID and the disentanglement scores (Fig. F.3). It can be seen that our model performs consistently better than the baseline, WGAN, regardless of the different $A$ values.

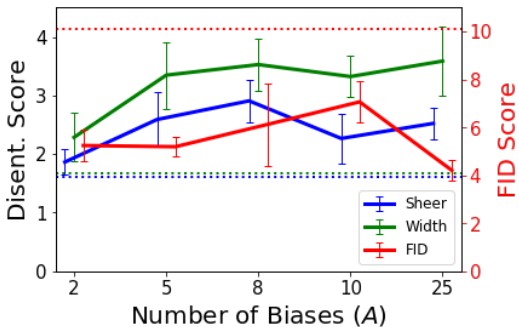

Figure F.3: The FID scores (right axis, the smaller the better) and the disentanglement scores (left axis, the larger the better) of MLA-GAN with varying $A$ are shown for MNIST dataset. The dashed lines show the mean scores of the baseline model (WGAN).

# G    EFFECT OF THE REGULARIZATION WEIGHT, $\lambda$, IN TRAINING

To investigate the effect of the regularization weight, $\lambda$, we train our model on MNIST with different $\lambda$ values. It can be seen that our model performs consistently better than the other compared models — MLA-GAN with no regularization ($\lambda = 0$), DMWGAN, MADGAN-like (see the next paragraph)

— regardless of the different $\lambda$ values (other models are omitted for better readability; see Table 1 for the omitted ones). It can be also seen that the scores are not very sensitive to the different choices of $\lambda$'s; this is beneficial in that one may choose any reasonable value for $\lambda$ when training the model with a new dataset.

Here, *MADGAN-like* is a DMWGAN model, but has a similar parametrization to the MADGAN (Ghosh et al., 2017): The parameters of the first three layers (from the top) are shared for all the generators. In contrast, MLA-GAN shares the parameters in all the layers except the biases in the fully-connected layers. Thus, in a sense, one can say that MLA-GAN shares only the last few layers, whereas *MADGAN-like* shares only the first few layers (of course, there is another difference due to the regularizers). Although the both models have the shared structures among the generators, MLA-GAN performs much better in both of the scores as seen in Figure G.4. This suggests that a simple parameter sharing is not enough to obtain a good performance, and the bias regularizer is indeed required.

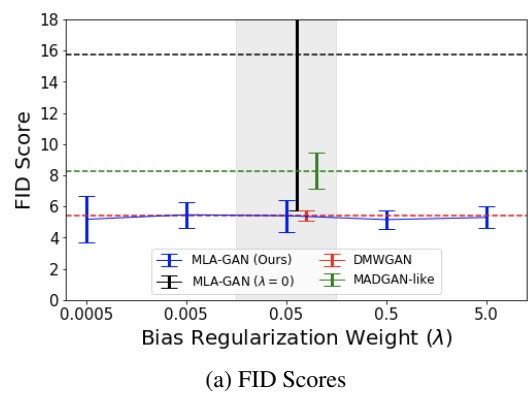

(a) FID Scores

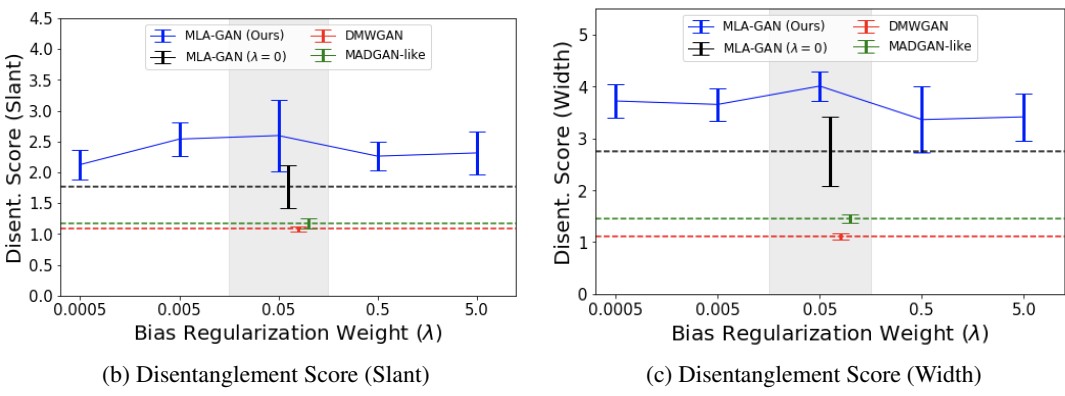

(b) Disentanglement Score (Slant)        (c) Disentanglement Score (Width)

Figure G.4: The FID scores (the smaller the better) and the disentanglement scores (the larger the better) of MLA-GAN with varying $\lambda$ are shown for MNIST dataset. Note the other models are positioned in the center (shaded in gray) to be visually comparable with the best-performing MLA-GAN model ($\lambda = 0.05$).

## H    EFFECT OF THE REGULARIZATION WEIGHT, $\mu$, IN ENCODING

To investigate the effect of the regularization weight, $\mu$, in encoding (Eq. 4), we take a trained model and encode an image from the train set using different $\mu$ values. Then, plugging in the encoded (estimated) biases, $\{a_l\}_{l=1}^{L}$, we randomly generate the samples from this new manifold and compare the qualities for different $\mu$'s.

From Figure H.5 (b), we can see that the quality of the encoding tends to improve as $\mu$ gets smaller. This might seem opposite to what we expect, as weaker regularization gives better results. However, looking closer, we can see that the features like the slant and the stroke are more aligned with a

stronger regularization of $\mu = 0.05$, when comparing with the other manifolds in the bottom pane. Thus, a trade-off exists here between the image quality of the samples and the alignment of the manifold to the others. Note we chose to use $\mu = 0.05$ in all the other experiments.

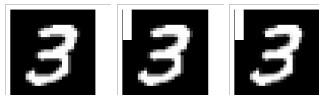

(a) **Left**: An image taken from the training dataset. **Middle**: The same image with a rectangle noise. **Right**: Regenerated image from the estimated biases, $\{a_l\}_{l=1}^{L}$, and the estimated latent code $z$, from Eq. 4.

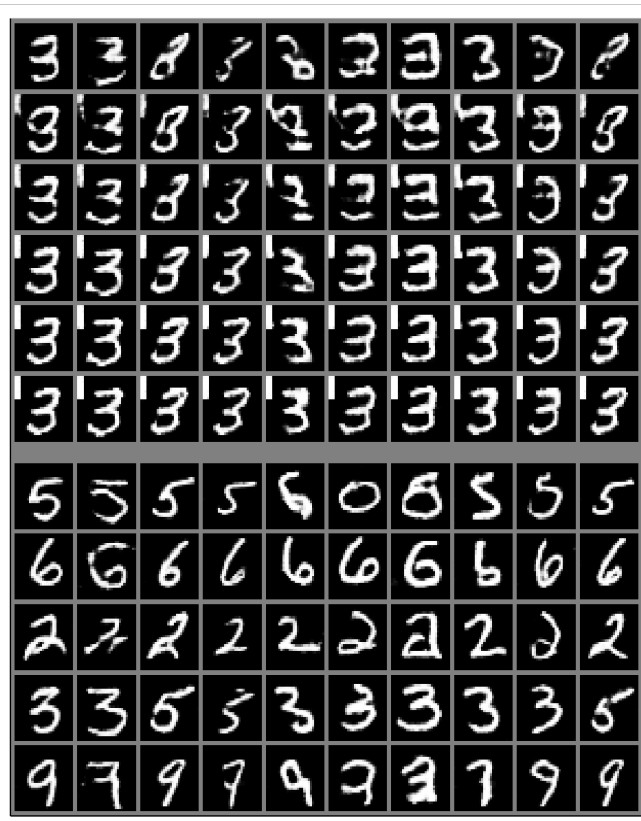

(b) **Top pane**: Generated samples from the estimated biases, $\{a_l\}_{l=1}^{L}$. The biases are estimated with different $\mu$'s, (5.0, 0.5, 0.05, 0.005, 0.0005, 0.0), from the top to the bottom. **Bottom pane**: Generated samples from the originally-learned biases, $\{\{a_l^{(i)}\}_{i=1}^{A}\}_{l=1}^{L}$ (only 5 out of 10 are shown). Note the images in the same column have the same latent value $z$.

Figure H.5: Effect of the regularization weight $\mu$ in encoding is shown with a MNIST-trained MLA-GAN model.

## I    EFFECT OF THE BIAS REGULARIZER

To examine the effectiveness of our bias regularizer, we visualize the raw values of biases $\{a_l^{(i)}\}_{i,l}$ and their (pseudo-)tangential component $\{U_l^{\top} a_l^{(i)}\}_{i,l}$ (see Fig. I.6, I.7). In all figures, we see that the biases are diverse, but their tangential components are well aligned due to the bias regularizer (left). On the contrary, without the regularizer, the tangential components are not aligned (right).

## J    SAMPLES GENERATED FROM VARIOUS MODELS

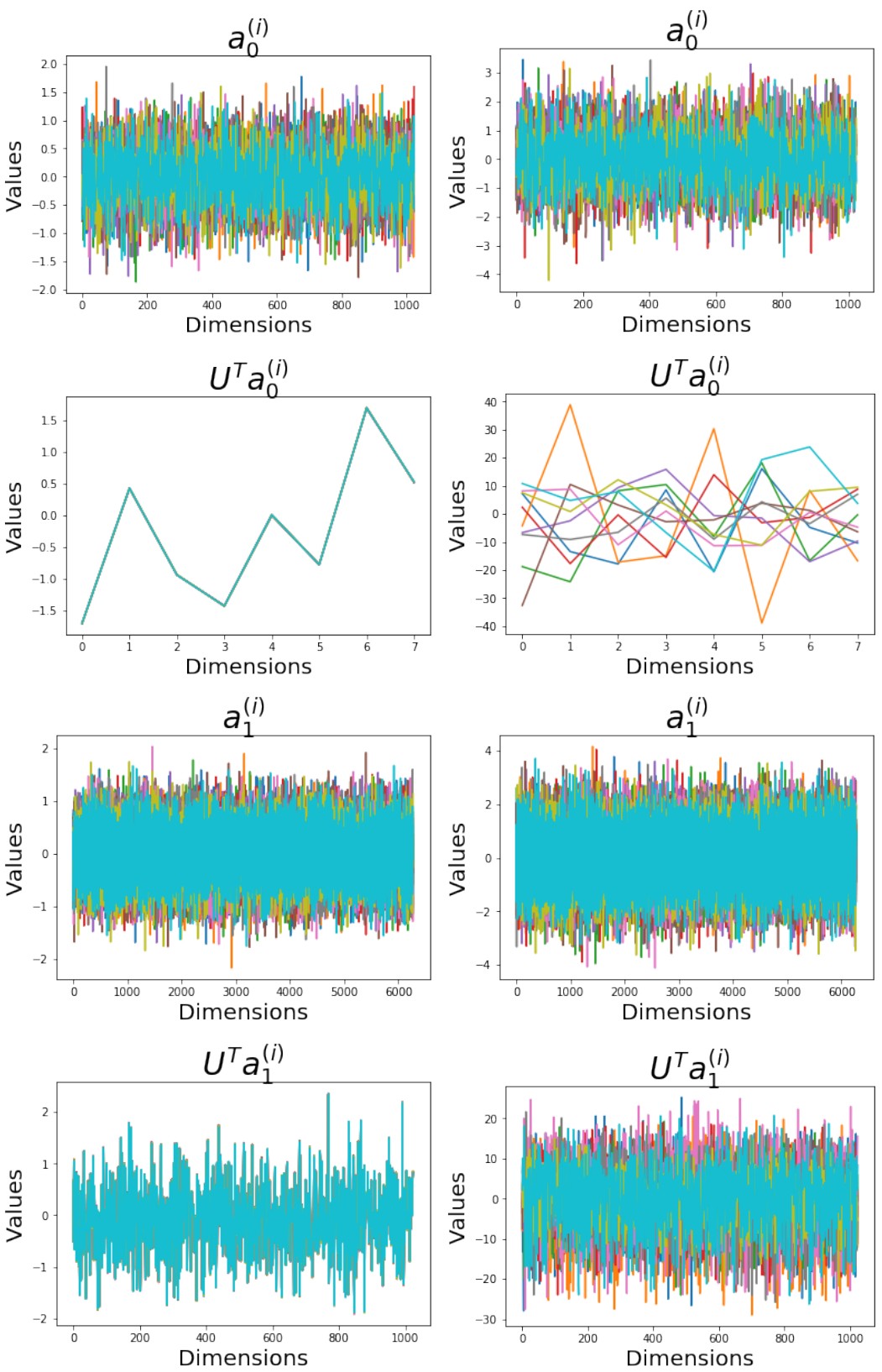

Figure I.6: Biases $a_l^{(i)}$ and their (pseudo-)tangential components $U_l^\top a_l^{(i)}$ of the MLA-GAN models, trained on ***MNIST***. Individual curve indicates each $i$-th bias. **Left** Parameters of MLA-GAN **Right** Parameters of MLA-GAN without the regularizer ($\lambda = 0$). It can be seen that the regularizer makes the tangential components of the biases well aligned.

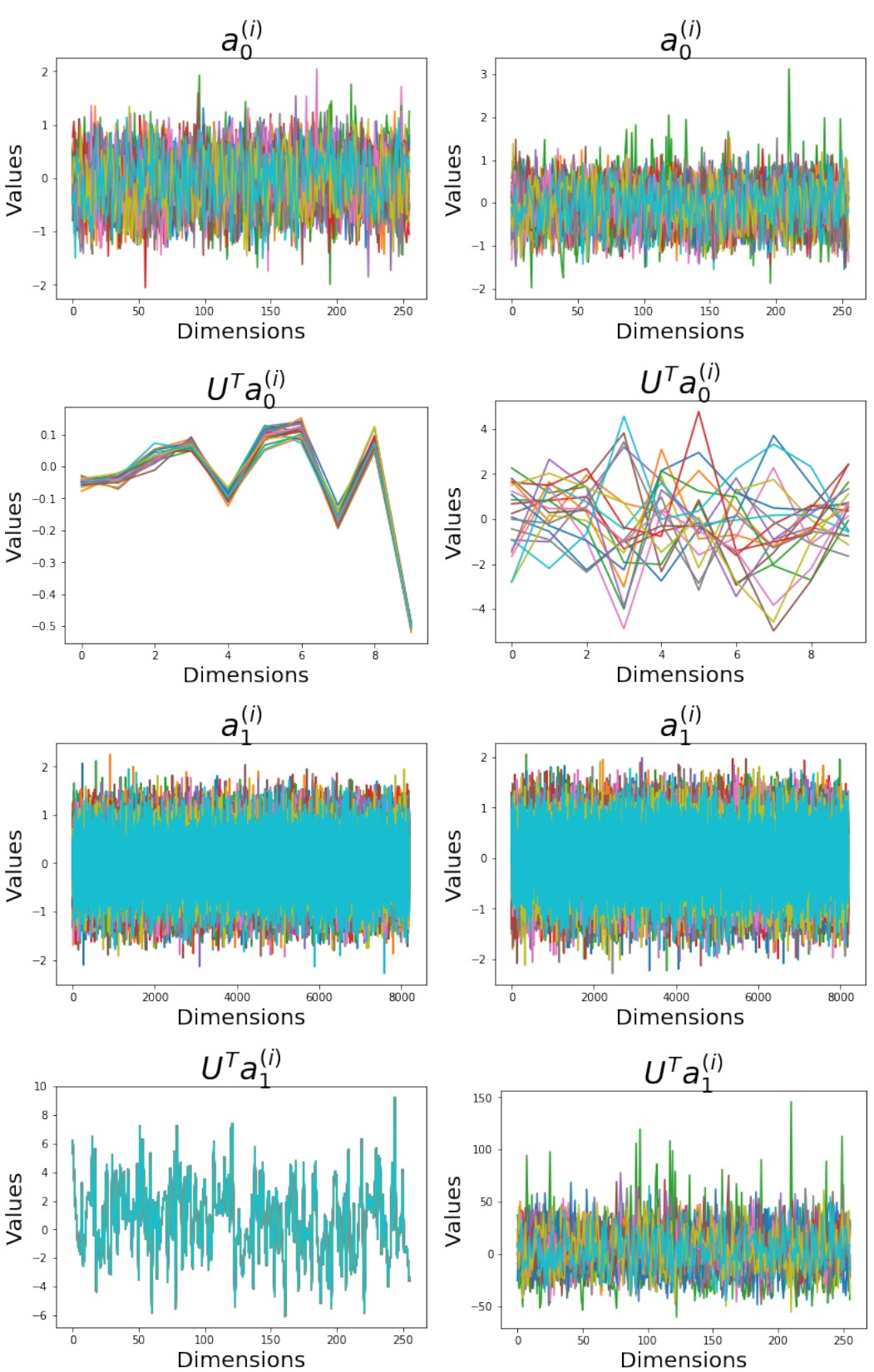

Figure I.7: Biases $a_l^{(i)}$ and their (pseudo-)tangential component $U_l^\top a_l^{(i)}$ of the MLA-GAN models, trained on **3D-Chair**. Individual curve indicates each $i$-th bias. **Left** Parameters of MLA-GAN **Right** Parameters of MLA-GAN without the regularizer ($\lambda = 0$). It can be seen that the regularizer makes the tangential components of the biases well aligned.

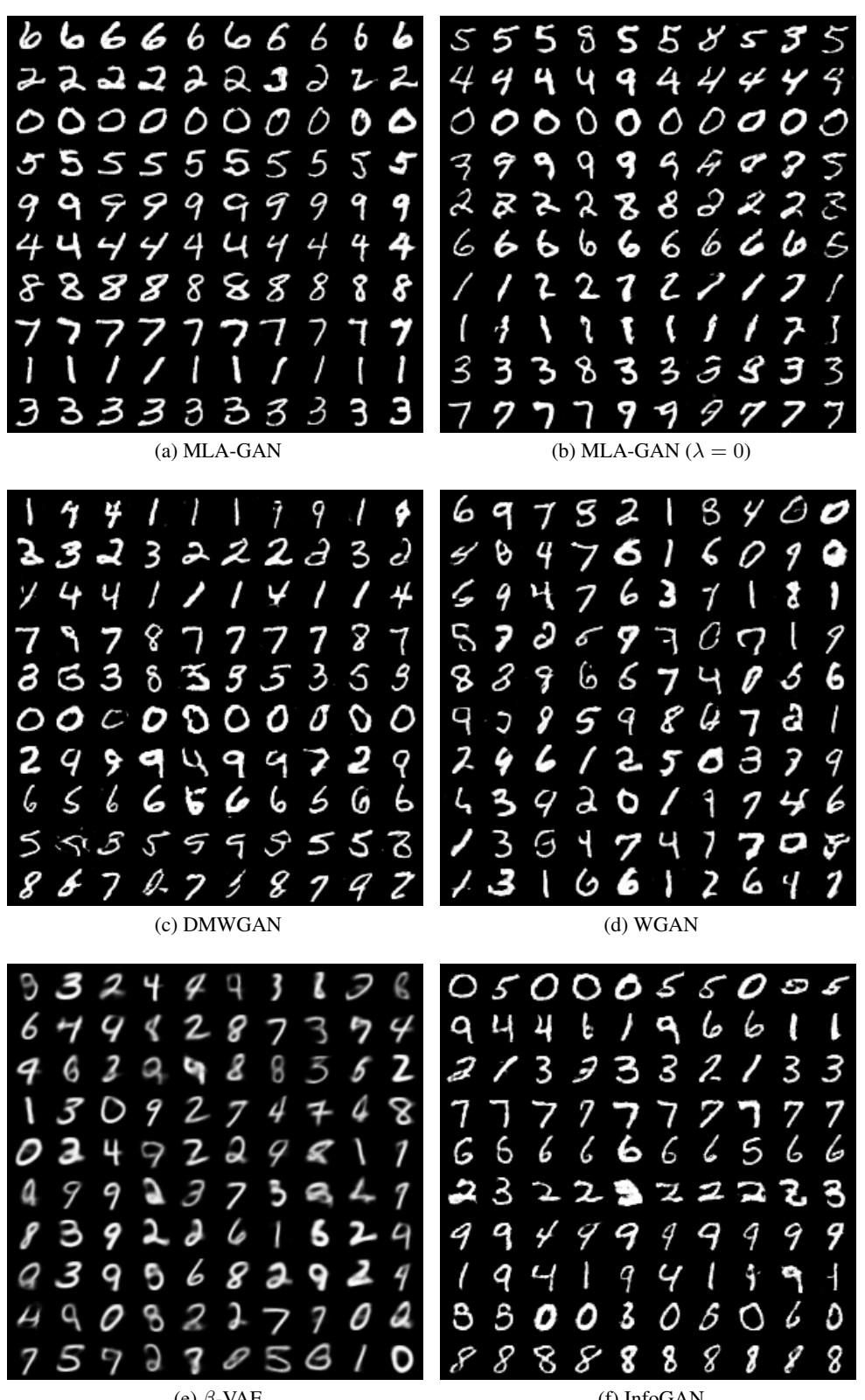

Figure J.8: MNIST image samples generated from the trained models

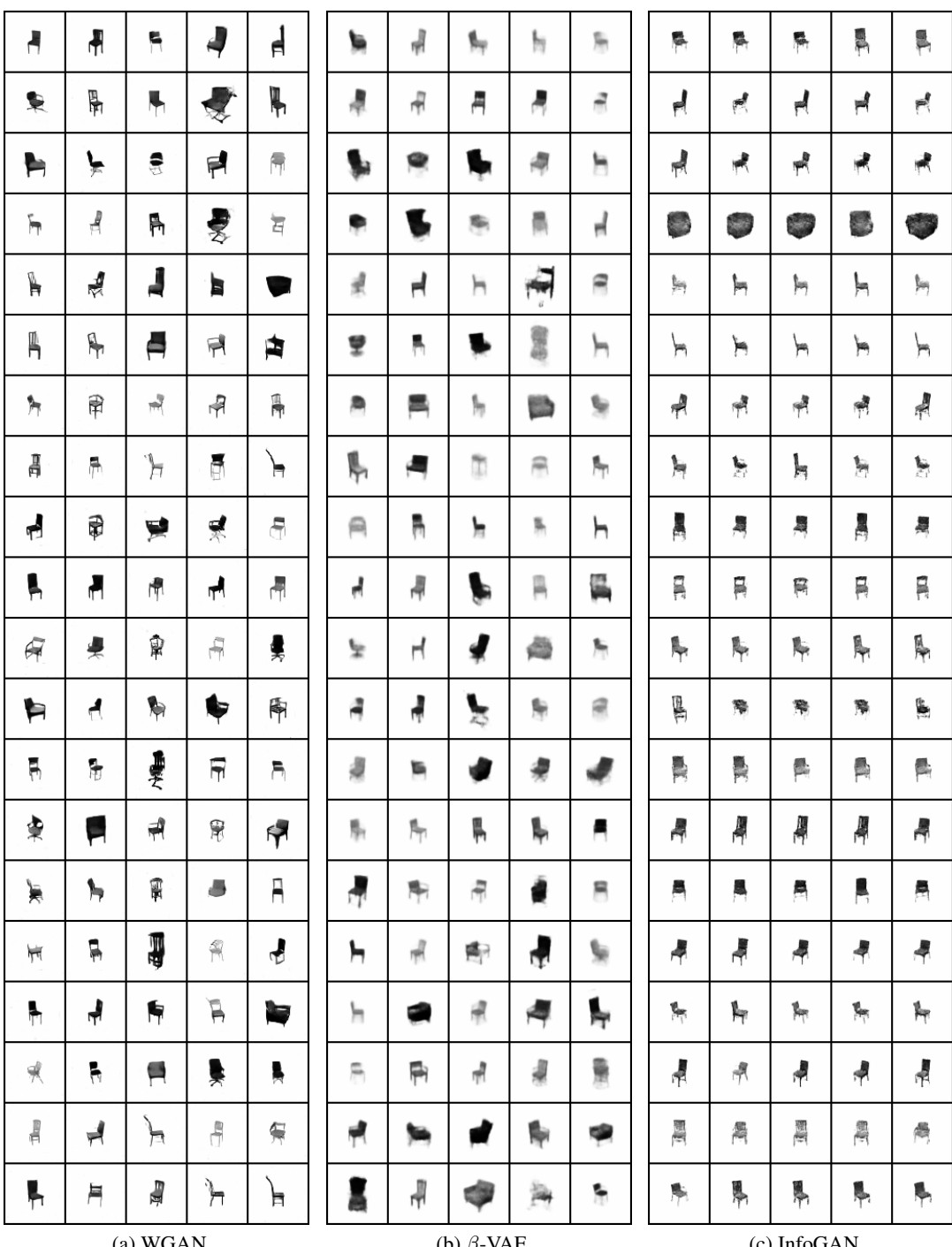

(a) WGAN      (b) $\beta$-VAE      (c) InfoGAN

Figure J.9: 3D-Chair image samples generated from the trained models

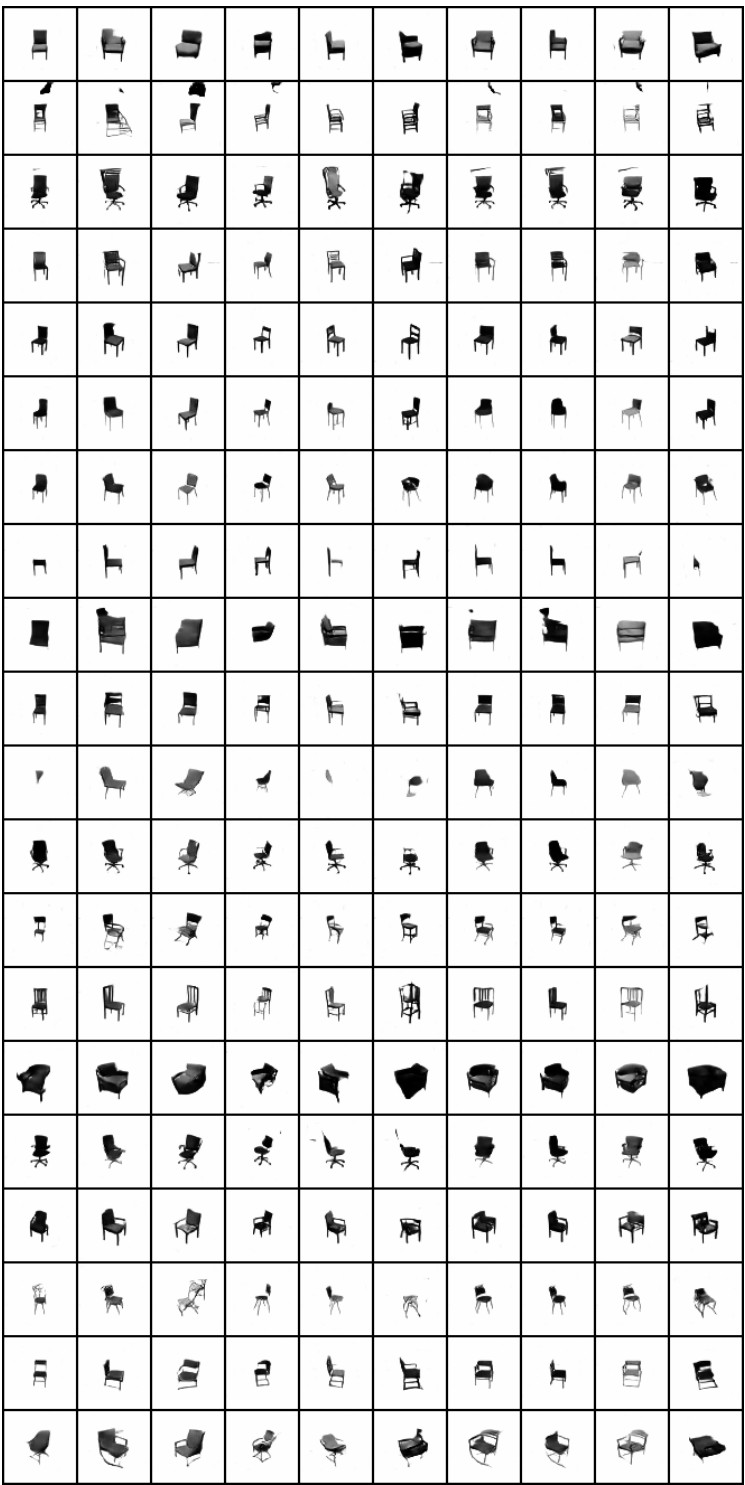

Figure J.10: 3D-Chair image samples generated from the trained MLA-GAN

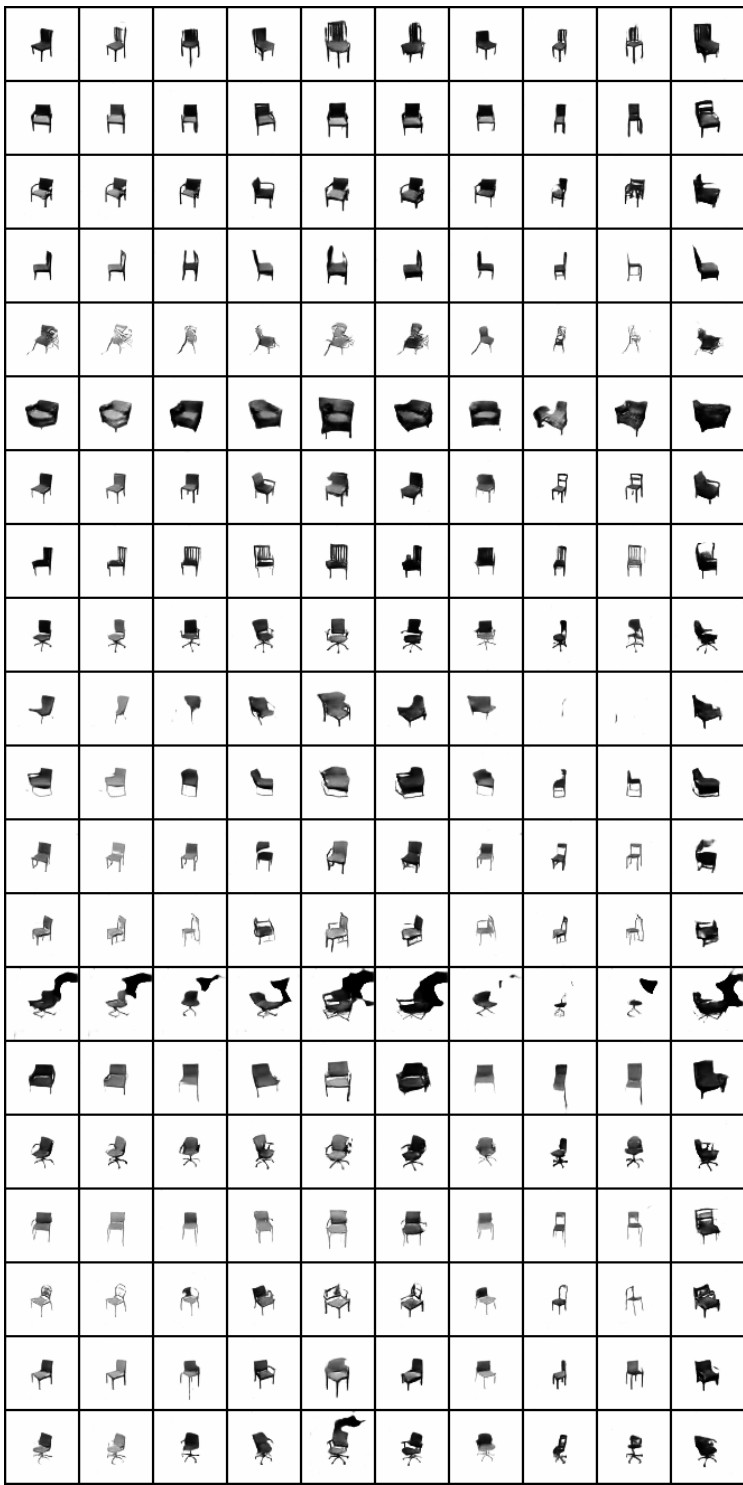

Figure J.11: 3D-Chair image samples generated from MLA-GAN ($\lambda = 0$)

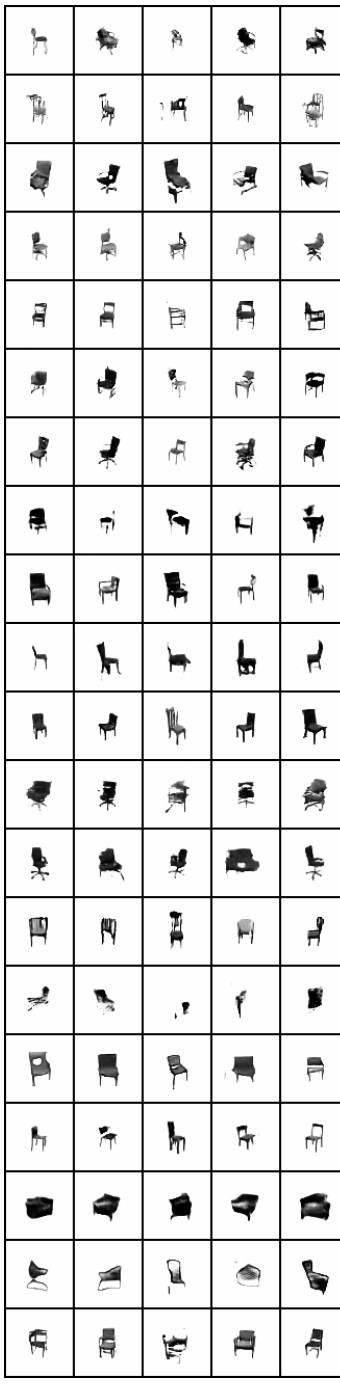

Figure J.12: 3D-Chair image samples generated from the trained DMWGAN

