# OpenReview forum: "Manifold Learning and Alignment with Generative Adversarial Networks"
_ICLR.cc/2020/Conference — Reject_

### Official Review · AnonReviewer2 · 2019-10-16
**Official Blind Review #2**

**Rating:** 6

**Review:**

A paper's idea is to train joint Wasserstein GAN for k datasets given in R^n (corresponding, e.g., to different classes of objects), together with manifold alignment of k manifolds.

The idea is to align points whose corresponding latent points are the same. This induces a natural constraint: k encoder functions (inverse of generators) should be consistent with each other. This is done by adding a regularization term. A paper demonstrates a clear motivation and a working solution for the problem. Experiments are convincing.

The only question is that the regularization term forces something stronger than just consistency of encoders. It seems, the requirement that "all  tangential components of biases are the same" means that k images of latent space (under k generator functions) are either coincide or non-intersecting. This is much stronger than just consistency, which is the weakest part of the approach.

**Experience Assessment:**

I have published one or two papers in this area.

**Review Assessment: Checking Correctness Of Derivations And Theory:**

I assessed the sensibility of the derivations and theory.

**Review Assessment: Checking Correctness Of Experiments:**

I assessed the sensibility of the experiments.

**Review Assessment: Thoroughness In Paper Reading:**

I made a quick assessment of this paper.

---

> ### Author Response · Authors · 2019-11-08
> **Thank you for your supportive comments.**
>
> Thank you for your supportive comments.
>
> You are correct about the strength of the regularization. Our formulation is indeed not the most general one that meets the consistency of encoders. However, we would like to note that this does not substantially restrict the expressiveness of our model as we use multiple layers. Although the top linear layer would only produce a set of parallel manifolds, the following nonlinear activation bends or folds the manifolds at different points. Processed similarly by the rest of the linear and nonlinear layers, the resulting manifolds in the bottom layer are much more complex than just being parallel to each other.
>
> Nevertheless, it is true that intersecting manifolds are hard to be represented by our model, as you pointed out. In practice, however, the intersecting manifolds are successfully learned in the form of manifolds with small gaps at the intersecting points. You can observe this in Zap50k results in Figure 3, the fourth column from the right, "the high heels." Although each of the four manifolds presents a different class of shoes, they seem to intersect near the high heels, and our model shows no difficulty in modeling such an intersection (except the intersection could have been approximated as a small gap).

---

### Official Review · AnonReviewer3 · 2019-10-23
**Official Blind Review #3**

**Rating:** 6

**Review:**

The paper suggests performing simultaneous manifold learning and alignment by multiple generators that share common weight matrices and a constructed inverse map that is instantiated by a single encoder. The method utilizes a special regularizer to guide the training. It has been empirically well tested on a multi-manifold learning task, manifold alignment, feature disentanglement, and style transfer.
Overall, this is an interesting idea with a motivated approach, however, I would like several points to be addressed before I could increase a score.
1. It seems that the method makes direct use of the number of classes in the datasets used. How would it fare compared to other models when the number of manifolds is not known (e.g. CelebA dataset)?
2. In MADGAN (Ghosh et al., 2017) generators share first layers which possibly makes them not independent as claimed in the paper, thus it is worth checking if MADGAN exhibits any kind of manifold alignment and could be a baseline for disentanglement with multiple generators.
3. There are hyperparameters \lambda and \mu for the regularizers in the model. It would be helpful to study their effect of different values on the training and encoding.
4. Is there a reason for DMWGAN/InfoGAN scores being omitted in Table 1?

Minor remark - there are a number of typos in the text.

**Experience Assessment:**

I have read many papers in this area.

**Review Assessment: Checking Correctness Of Derivations And Theory:**

I assessed the sensibility of the derivations and theory.

**Review Assessment: Checking Correctness Of Experiments:**

I assessed the sensibility of the experiments.

**Review Assessment: Thoroughness In Paper Reading:**

I read the paper thoroughly.

---

> ### Author Response · Authors · 2019-11-08
> **Thank you for your supportive comments.**
>
> Thank you for your supportive comments. We would like to address the points one by one.
>
> 1. We agree that the number of generators ($A$) is one of the crucial factors in modeling. In fact, the number of classes in the dataset is not the optimal number, but only a reasonable number for $A$. In this regard, we conducted additional experiments with MNIST using different $A$'s, as shown in Figure F.3. It can be seen that our model, regardless of the different $A$ values, performs consistently better than the baseline WGAN model. Interestingly, $A=10$ was not the best setting for MNIST; it was $A=25$. This suggests a need for learning $A$ from data, but we think this is beyond our current scope, as discussed in Sec. 2.3.2.
>
> 2. This is a very good point. Although the original purpose of sharing weights in MADGAN was to avoid redundant computations, it is definitely worth checking if this design contributes to the manifold alignment. We are working on reproducing the MADGAN experiments, and we will report the results in a few days.
>
> 3. Good point. Note the current version already includes the results for $\lambda=0$ (see Table 1 and Appendix G). We will add the results for different $\lambda$'s and $\mu$'s soon.
>
> 4. As the generators of DMWGAN are not at all correlated to each other, we initially thought that showing the manifold-alignment performance of DMWGAN makes little sense. We will report these scores, both for MNIST and 3D-Chair, in a couple of days (InfoGAN scores are filled now).
>
> Typos: We have fixed the typos, if not all. We will review the text more thoroughly and revise it before submitting the final version.

---

> > ### Author Response · Authors · 2019-11-15
> > **We conducted the suggested experiments and added the results.**
> >
> >
> > Dear R3,
> >
> > We conducted several additional experiments you suggested and updated the manuscript with the results.
> >
> > ------------------------------------------------------------------------------------------
> > Please see
> > - Appendix F, regarding your comment#1 (as we have already answered in our previous comment).
> > - Appendix G, regarding your comment#2 (manifold alignment performance of MADGAN) and comment#3 (the effect of using different $\lambda$'s).
> > - Appendix H, regarding your comment#3 (the effect of using different $\mu$'s).
> > - Updated Table 1, regarding your comment#4.
> > ------------------------------------------------------------------------------------------
> >
> > Brief summary of what we found from the additional experiments:
> >
> > - Using different $\lambda$'s did not significantly affected the performance.
> > - Using different $\mu$'s brings up a trade-off situation between the sample quality (manifold estimation quality) and the manifold alignment quality. The value we used was around the middle of the two extremes.
> > - Instead of MADGAN, we tested a (MADGAN-like) DMWGAN model where the parameters of the first three layers are shared among the generators. This model showed a bit better performance than DMWGAN, but far worse than the MLA-GAN.

---

### Official Review · AnonReviewer1 · 2019-10-26
**Official Blind Review #1**

**Rating:** 6

**Review:**

EDIT: Updated score to weak Accept in lieu of author's response.  See below for more details.

The authors propose a GAN architecture that aims to align the latent representations of the GAN with different interpretable degrees of freedom of the underlying data (e.g., size, pose).  While the text, motivation, and experiments are fairly clear, there are some spelling/grammar mistakes throughout, and the draft could use a solid pass for overall clarity.

While the idea of using the log trace covariance as a regularizer for the manifold is certainly interesting, it seems fairly incremental upon previous work (i.e., DMWGAN).  Even modulo the work being incremental, I still have concerns regarding the comparison to baselines/overall impact, and thus I suggest a Weak Rejection.

Table 1 seems to indicate that the author's proposed method is on par with or worse than every method compared against except for 3D chair (bright).  Additionally, the lack of comparison against DMWGAN for every task (except the first) is a bit concerning, considering its similarity to the proposed method.  If the authors could check DMWGAN's performance for all of their tasks and report it, I would be more likely to raise my score.

**Experience Assessment:**

I have read many papers in this area.

**Review Assessment: Checking Correctness Of Derivations And Theory:**

I assessed the sensibility of the derivations and theory.

**Review Assessment: Checking Correctness Of Experiments:**

I assessed the sensibility of the experiments.

**Review Assessment: Thoroughness In Paper Reading:**

I read the paper at least twice and used my best judgement in assessing the paper.

---

> ### Author Response · Authors · 2019-11-08
> **Thank you for your valuable comments.**
>
> Thank you for your valuable comments.
>
> We believe your concerns arose mainly from the quantitative results in Table 1. We would like to resolve the concerns by analyzing Table 1 in detail, but before that, we invite you to see the qualitative results from Figure H.6 in Appendix H for the better discussion.
>
> By comparing (a) and (c) of Figure J.8, we can clearly see the superior performance of MLA-GAN over DMWGAN (Note the samples are arranged in the same manner as Figure 3.). Column-wise, we see the samples share the same smooth features (e.g., stroke, slant) in MLA-GAN, but this is not the case at all in DMWGAN. Row-wise, we see the generators of MLA-GAN present distinct digit manifolds with a clean separation, whereas the generators of DMWGAN present manifolds involving a few crossovers between the digits; this is likely because the generators of MLA-GAN are enforced to share the smooth features, driving more regular manifold structures in all the generators.
>
> From these clear differences and benefits, we disagree that MLA-GAN is incremental to DMWGAN. Most of all, our principal objective was not only the multi-manifold learning but also the manifold alignment, but DMWGAN cannot perform the manifold alignment, as illustrated in Figure 1 and discussed in Section 3. Also, we would like to emphasize that the generalizability of MLA-GAN to an untrained manifold, demonstrated in the style-transfer experiments, is another very distinct property of MLA-GAN over other models.
>
> That being said, we agree that Table 1 is lacking information about DMWGAN. We are currently running the experiments, and we assure you that all the unfilled scores will be reported in a couple of days (The reason that we did not report the disentanglement scores at first was that the DMWGAN has almost nothing to do with the manifold alignment).
>
> Your last concern about Table 1 was that the disentanglement scores of our model are inferior to that of $\beta$-VAE. But as we have pointed out in Section 4.2, the manifold learning performance (FID) of $\beta$-VAE is far worse than our model (see also, Figure J.8 (e) and Figure J.9 (b)). We emphasize that the FID and the disentanglement scores should be simultaneously considered to evaluate the MLA task, and MLA-GAN is showing the best performance in that regard.
>
> Typos and grammatical errors: We have fixed the most, if not all. We will review the text more thoroughly and revise it before submitting the final version.
>
> == Minor Edit ==
> We updated some of the appendix-figure indices in this comment, since they are changed as we add more figures in the revised manuscript.

---

> > ### Comment · AnonReviewer1 · 2019-11-15
> > **Updating score**
> >
> > I greatly appreciate the effort by the authors to further contextualize their results, as well as provide updates to their table.  I've increased my score to 6 in light of this.
> >
> > Regarding incrementality, it seems the original DMWGAN authors proposed using something quite similar to the author's proposed regularizer (without the log)--i.e., the DMWGAN authors monitor Tr[Cov[x]] as a quality measure, and the present manuscript optimizes Log[Tr[Cov[x]]].  While I agree that clearly nontrivial effort went forth into justifying this choice of regularizer--could the authors comment on the similarity?

---

### Author Response · Authors · 2019-11-13
**The scores of DMWGAN is added in Table 1**

Dear reviewers

As suggested by R1 and R3, we just filled in the scores of DMWGAN in Table 1 (please see the updated manuscript). Note that these scores are obtained from the best performing setting among we have examined, to be fair to DMWGAN; we tried adding/removing the BatchNorm layers and using different division numbers (which effectively changes the number of hidden units in every layer). The detailed experimental setting can be found in Appendix C.

To briefly discuss the added scores, we can first see that the disentanglement scores of DMWGAN are very close to the base value (which is one), being the worst among the other models. This is the expected result, however, since DMWGAN uses a set of uncorrelated generators which gives no meaningful alignment in the latent space (see Figure 1, middle, as a reminder). Secondly, we can see that the FID score of DMWGAN in 3D-Chair dataset is also quite bad. We carefully speculate that the effectiveness of MI regularizer in DMWGAN has been saturated here, since it becomes hard to assign each generator to each datum as the dataset gets complicated.

We will update the results of the other experiments suggested by R3 very soon (the effect of weight sharing in the first few layers as in MADGAN; the effect of regularization weights $\lambda$, $\mu$).

---

### Decision · Program_Chairs · 2019-12-19

**Decision:**

Reject

**Comment:**

This work proposes a GAN architecture that aims to align the latent representations of the generator with different interpretable degrees of freedom of the underlying data (e.g., size, pose).

Reviewers found this paper well-motivated and the proposed method to be technically sound. However, they cast some doubts about the novelty of the approach, specifically with respect to DMWGAN and MADGAN. The AC shares these concerns and concludes that this paper will greatly benefit from an additional reviewing cycle that addresses the remaining concerns.